# CARS Imaging Advances Early Diagnosis of Cardiac Manifestation of Fabry Disease

**DOI:** 10.3390/ijms23105345

**Published:** 2022-05-11

**Authors:** Elen Tolstik, Nairveen Ali, Shuxia Guo, Paul Ebersbach, Dorothe Möllmann, Paula Arias-Loza, Johann Dierks, Irina Schuler, Erik Freier, Jörg Debus, Hideo A. Baba, Peter Nordbeck, Thomas Bocklitz, Kristina Lorenz

**Affiliations:** 1Leibniz-Institut für Analytische Wissenschaften-ISAS-e.V., Bunsen-Kirchhoff-Str. 11, 44139 Dortmund, Germany; p.ebersbach@exeter.ac.uk (P.E.); johann.dierks@isas.de (J.D.); irina.schuler@isas.de (I.S.); erik.freier@uni-wuppertal.de (E.F.); 2Leibniz Institute of Photonic Technology, Member of Leibniz Health Technologies, Albert-Einstein-Str. 9, 07745 Jena, Germany; nairveen@hotmail.com (N.A.); shuxia_guo@seu.edu.cn (S.G.); thomas.bocklitz@leibniz-ipht.de (T.B.); 3Institute of Pathology, University Hospital Essen, Hufelandstr. 55, 45147 Essen, Germany; dorothe.moellmann@uk-essen.de (D.M.); hideo.baba@uk-essen.de (H.A.B.); 4Department of Nuclear Medicine, Oberdürrbacher Str. 6, 97080 Wuerzburg, Germany; arias_p@ukw.de; 5Department of Physics, TU Dortmund University, Otto-Hahn-Str. 4a, 44227 Dortmund, Germany; joerg.debus@tu-dortmund.de; 6Department of Internal Medicine I, University of Würzburg, Oberdürrbacher Str. 6, 97080 Wuerzburg, Germany; nordbeck_p@ukw.de; 7Institute of Pharmacology and Toxicology, University of Würzburg, Versbacher Str. 9, 97078 Wuerzburg, Germany; 8Comprehensive Heart Failure Center, University Hospital of Würzburg, Am Schwarzenberg 15, 97078 Wuerzburg, Germany

**Keywords:** coherent anti-Stokes Raman scattering (CARS) microscopy, Raman micro-spectroscopy, cardiovascular diseases, Fabry Disease (FD), Gb3 and lyso-Gb3 biomarkers, multivariate data analysis, immunohistochemistry

## Abstract

Vibrational spectroscopy can detect characteristic biomolecular signatures and thus has the potential to support diagnostics. Fabry disease (FD) is a lipid disorder disease that leads to accumulations of globotriaosylceramide in different organs, including the heart, which is particularly critical for the patient’s prognosis. Effective treatment options are available if initiated at early disease stages, but many patients are late- or under-diagnosed. Since Coherent anti-Stokes Raman (CARS) imaging has a high sensitivity for lipid/protein shifts, we applied CARS as a diagnostic tool to assess cardiac FD manifestation in an FD mouse model. CARS measurements combined with multivariate data analysis, including image preprocessing followed by image clustering and data-driven modeling, allowed for differentiation between FD and control groups. Indeed, CARS identified shifts of lipid/protein content between the two groups in cardiac tissue visually and by subsequent automated bioinformatic discrimination with a mean sensitivity of 90–96%. Of note, this genotype differentiation was successful at a very early time point during disease development when only kidneys are visibly affected by globotriaosylceramide depositions. Altogether, the sensitivity of CARS combined with multivariate analysis allows reliable diagnostic support of early FD organ manifestation and may thus improve diagnosis, prognosis, and possibly therapeutic monitoring of FD.

## 1. Introduction

The establishment of novel laser diagnostic techniques for faster, precise, and more sensitive detection of disease-associated changes [1,2,3,4,5] has gained increasing attention for improving early diagnosis [6]. Hence, alternatives to standard diagnostics are of great clinical need [7]. One of the diseases that would profit from earlier diagnosis is Fabry Disease (FD) [8,9].

FD is an X-linked disorder of the glycosphingolipid metabolism due to mutations within the α-galactosidase A (α-Gal A) gene that leads to reduced enzyme activity and subsequently to progressive accumulation of neutral glycosphingolipids, especially globotriaosylceramide (Gb3) and its metabolite globotriaosylsphingosine (lyso-Gb3) in lysosomes, endoplasmic reticulum, in the cytoplasm and the nucleus, as well as to increased secretion of Gb3 and lyso-Gb3 [6,10,11,12,13,14,15,16]. FD is considered a multi-organ disease with severe clinical symptoms, particularly regarding the heart, kidneys, brain, and the central nervous system [6,9,12,17]. Cardiac involvement is common and represents the most prevalent cause of death in FD patients and thus carries the highest prognostic impact [14,18,19]. FD patients are categorized into two groups, classic and non-classic FD patients, based on their clinical presentation. The classic phenotype manifests during childhood and adolescence, affects males earlier than females, and includes multiple symptoms (e.g., pain, decreased or absent sweat production, discomfort in warm temperatures, the appearance of a reddish to dark-blue skin rash, and gastrointestinal problems) [20]. The non-classic phenotype manifests in the fourth to sixth decade of life. The heart is more frequently affected in this variant; these patients develop progressive left ventricular hypertrophy and myocardial fibrosis. In patients carrying certain Fabry mutations and female heterozygous carriers, the heart can even be the only affected organ [21]. In heterozygous female carriers, this can be explained by random X-chromosomal inactivation, which can be cell-type dependent and can lead to mosaic inactivation [15,18]. Thus, α-Gal A activity measured in blood or white blood cells cannot entirely exclude non-classic FD [22]. Also, it is unclear whether α-Gal A activity assessed in-vitro correlates with its substrate utilization in the respective organs and whether a critical threshold of enzyme activity in the different organs is needed to avoid Gb3 accumulations [20]. Taken together, the diagnosis of cardiac involvement in FD is very challenging. Effective pharmacological treatments are available, including enzyme replacement and chaperone therapy. In principle, both treatment options can hold, slow, or even partially reverse the disease progression, but they are only effective if FD is diagnosed at an early disease stage before irreversible fibrotic remodeling has occurred [9,12,15,23].

An impressive toolbox of methods has been applied for the diagnosis of cardiac involvement of FD. These methods include positron emission tomography (PET) and gadolinium-based contrast cardiac magnetic resonance imaging (MRI) [24,25], advanced echocardiography [20], and Doppler tissue imaging [24,26], as well as immunostaining and electron microscopy [27] to assess cardiac function and morphology, structural changes within the tissue, or substrate depositions. All these methods have certain limitations as they, for example, only monitor the phenotype, lack spatial resolution or sensitivity, or are very time-consuming. Thus, novel biomarkers and/or new diagnostic methods for the cardiac involvement of FD are urgently needed.

Here, we assess the use of Coherent anti-Stokes Raman scattering (CARS) in FD diagnosis. This label-free vibrational microscopy is based on the energy levels of the present atomic composition of molecules and thus possesses high specificity [28,29]. It is applied to measure the biochemical compositions of biological samples, their organization, and dynamics [7,30,31,32,33,34,35]. In this context, strong CARS signals of the aliphatic CH-stretching vibrations, i.e., Raman resonances at 2850 cm^−1^ and 2940 cm^−1^, provide important insights into the lipid/protein tissue composition [2,4,30,36]. In addition, CARS enables the imaging and localization of lipid-based compounds with high contrast due to a significant amount of CH-bonds [1]. Compared to other label-free vibrational microscopy methods such as Raman micro-spectroscopy, CARS microscopy produces strong signals (up to 10^6^–10^7^ times stronger than classical confocal Raman spectroscopy) and thus can be performed in much shorter periods, making it attractive for clinical application [5]. Due to near-IR irradiation, CARS imaging allows a high penetration depth of light in biological tissue with minor autofluorescence [37]. The combination of CARS with Second-Harmonic Generation (SHG) is advantageous since non-centrosymmetric structures such as collagen are capable of emitting SHG light so that additional morphological information can be retrieved from tissue biopsies [5]. To translate such morphological and biomolecular information to reliable diagnostics, multivariate statistical analysis algorithms and machine learning (ML) techniques must be applied. Here, we implemented an ML algorithm for CARS-SHG microscopy to detect an early cardiac involvement of FD with high mean sensitivity in an FD mouse model.

## 2. Results

Strong vibrational signals of lipids with high content of CH_2_ stretches have defined CARS microscopy as a suitable technique to visualize lipid-rich regions [5]. Here, we evaluate whether CARS imaging holds the potential to support the diagnosis of the lipid disorder FD, in particular, the detection of its cardiac manifestation. To study the application of CARS microscopy in FD, we used a well-characterized FD mouse model, the α-Gal A knockout mice (GLA^KO^), and compared it to GLA^WT^ mice with no α-Gal A gene deletion [26]. The hearts were analyzed at an early point (20 weeks of age) when the phenotypic changes are still rather non-specific in the heart as the cardiac deposition of Gb3 is very low and cardiac function largely unaffected [38,39]; a condition in which FD-specific therapy has the highest chance of success.

The workflow of the study is shown schematically in Figure 1. The goal of the project was the development of a differentiation method for cardiac biopsies based on a large database. For this study, cardiac sections of GLA^KO^ and GLA^WT^ mice were prepared, and 4–6 areas from each tissue section (defined as “image” in the following) were randomly selected for the assessment using a CARS-SHG microscope. To assess the biomolecular content and spectroscopic response of FD samples, hyperspectral scans of random areas within the biopsies and whole heart sections were recorded (see Section 4, Materials and Methods for detailed information). First, CARS-SHG imaging of the cardiac sections was performed by tuning the pump laser on 816.5 nm to measure particularly aliphatic CH_2_ stretching vibrations, which represent the dominant signals in lipids [40,41]. In Figure 2, representative stitched images of tile scans for both knock-out and wild-type mice (first and second row) along with a single image of a GLA^KO^ heart section (third row) are shown. The CARS channel (depicted in red) provides CH stretching vibrations of aliphatic side chains, while the SHG channel (depicted in green) is used to detect filamentous proteins in heart muscle cells. The merged images of the CARS and the SHG channel (right column) show a strong dominance of the CARS signal (depicted in red) in the tissue of GLA^KO^ hearts compared to the wild-type one. The CARS-SHG signals are evenly distributed throughout the tissue sections of the respective genotype. Even though the SHG-CARS mode already suggests significant differences between the genotypes, further computational analyses are needed to reliably and numerically differentiate between both genotypes.

For this, we recorded hyperspectral (xyΛ)-scans of random areas within the heart sections of both genotypes. The study includes xyΛ-heart section scans of 20 GLA^WT^ and 35 GLA^KO^ mice. The spectral range of 2702.2 cm^−1^ to 3039.3 cm^−1^ for CH stretching vibrations of aliphatic side chains was covered by varying lambda (Λ) between 804.0 nm and 826.4 nm. Based on these measurements, a large number of spectra were collected. To classify those and to be able to compare the different spectra recorded throughout the different tissue slices and genotypes, the spectra were “clustered” using the unsupervised k-means algorithm (see Section 4, Materials and Methods), which searches for groups of data points that have particular similarities. The most common clusters throughout all measurements for both genotypes are represented in Figure 3. Mean spectra of each k-means cluster were calculated for both genotypes. The most frequent clusters are shown for each genotype as spectral representation (Figure 3b,d) and as a histogram with proportional (%) distribution of the clusters in the different genotypes (Figure 3a,c). These results are presented to analyze three (Figure 3a,b) and four (Figure 3c,d) k-mean clusters. Indeed, the mean spectra of the three most frequent clusters were already sufficient different to separate the two groups and to demonstrate a clear differentiation between the genotypes based on the two main Raman shifts: the CH_2_ symmetric stretching at ~2850 cm^−1^ depicting lipids and the asymmetric CH_3_ stretching at ~2930 cm^−1^ depicting mainly proteins [1,42,43]. For GLA^WT^, the clustering resulted in an almost equal “protein-lipid ratio” presented as a yellow (Figure 3a) or a light blue (Figure 3c) cluster. For GLA^KO,^ in contrast, a superiority in lipid-rich regions was observed in tissue at ~2850 cm^−1^ illustrated as pink (Figure 3a) or pink/orange clusters (Figure 3c). Also, the calculation with up to eight clusters confirmed these results (not shown).

For the visual presentation of our results in the tissue sections, the respective spectra were reconstructed as color maps for the above-mentioned three and four clusters (Figure 3b,d, respectively). These color maps resulted in well-defined visual discrimination between the genotypes. For example, the “four cluster map” (Figure 3d) showed a predominant “equal protein-lipid” signal (cluster 4, light blue) in GLA^WT^. Contrary, GLA^KO^ heart tissues revealed a high “lipid” signal (cluster 3, pink). The application of the higher number of clusters (three versus four in Figure 3b,d) seems to enable an even higher differential resolution of the biochemical composition of tissue (e.g., yellow/pink ratio in Figure 3d). Thus, the visual presentation seems to be able to differentiate between the two genotypes but also contributes information on the spatial resolution of different clusters in FD biopsies.

Our results correlate well with the lipid depositions reported in the cardiac tissue of FD patients, i.e., the more prominent lipid signal in GLA^KO^ compared to wild-type mice. To substantiate the data statistically, we combined the principal component analysis (PCA) algorithm with the linear discriminant analysis (LDA) model. The LDA classification finds optimal linear combinations of variables intending to maximize the differences between two genotypes and, at the same time minimizing the differences within one group [7]. These modeling techniques consider the influence of biological variations on genotype differentiation in their analyses. Along with the PCA-LDA classifier, a leave-one-mouse-out cross-validation (LOMO-CV) was applied to predict the genotypes based on Gb3 associated depositions and to study the reproducibility of the LDA model [44]. To further consider the biological variability within tissue sections, this classification model was applied to selected data sets as described in the following for “Datasets I–III” (Figure 4). Shown are three exemplary analyses: (I) 20 images of GLA^WT^ vs. 35 of GLA^KO^ extracted from 9 GLA^WT^ and 10 GLA^KO^ mice (“more images per mouse”; Dataset I); (II) 10 images of GLA^WT^ vs. 28 of GLA^KO^ extracted from 7 GLA^WT^ and 7 GLA^KO^ mice (“an equal number of mice per genotype”; Dataset II); and (III) a data set with only one image per mouse, i.e., 8 GLA^WT^ vs. 10 GLA^KO^ (“one image per mouse”; Dataset III).

The mean spectra of the two genotypes within the chosen datasets (I–III) are shown in Figure 4 (Ia, IIa and IIIa). Indeed, the mean spectra of GLA^KO^ and GLA^WT^ showed clear differences for all of these datasets. In all GLA^KO^ groups, the most prominent CARS signal was around 2850 cm^−1^ (Figure 4 (Ia, IIa and IIIa)), which resembles the “lipid-associated” peak. GLA^WT^ samples, however, show a double peak with similar signal/peak strength resembling the CH_2_ and CH_3_ stretching modes for lipids and proteins, respectively (Figure 4 (Ia, IIa and IIIa)). Thus, the mean spectra suggest a separation between the genotypes. A method that is frequently applied to analyze the discrimination of groups, the PCA scattering plots (biplot), only revealed a slight separation of the genotypes (Figure 4b). The biplot was applied for the first two PCs (PC1 and PC2) for all three datasets (groups I–III), where each dot represents a respective data point of the different genotypes and is depicted in green or orange for GLA^WT^ or GLA^KO^ mice, respectively.

To further analyze and reliably interpret the data, classification models were applied. The classification results using a PCA-LDA (described in detail in Section 4, Materials and Methods) for the increasing number of PCs for both genotypes are plotted in Figure 4c. The PCA-LDA combinations for each data set (I–III) based on leave-one-mouse-out cross-validation (LOMO-CV) result in a clear genotype separation between the studied groups (presented in Figure 4c). For the analyses, an LDA model for each data set was constructed using increasing numbers of PCs (1, 2,…,14) of all images of all mice except for the ones of the chosen mouse, of which only one fixed image was selected. This procedure was repeated for each image of all mice. Each of these calculations resulted in predicting one or the other genotype. The mean sensitivity of the predicted results was calculated. It exceeded 85% for all studied groups when at least two PCs were used for the LDA model. The selected data sets showed maximal mean sensitivities of 90.71% (I), 96.43% (II), and 95% (III) for the indicated numbers of PCs (marked as red dots).

Data sets I and II required at least 9 or 12 PCs, respectively, to separate the groups with high classification mean sensitivity >90% via LDA. In dataset III, a high classification sensitivity had already been achieved with only 2 PCs. Apparently, the spectral variations in dataset III (“one image per mouse”) are more dominated by group-specific spectral features, while the group-specific spectral features in datasets I and II hardly affect the spectral variations and are thus only visible with a higher number of PCs. Taken together, the different analyses of the spectra that were applied resulted in different discrimination strengths regarding the two genotypes.

To further compare the sensitivity of CARS measurements and the applied bioinformatic evaluation with a standard method, we performed immunohistochemical (IHC) using anti-Gb3 antibodies [27]. As mentioned above, this method is commonly used to support the FD patient diagnosis regarding specific organ involvement. Gb3 accumulations in lysosomes are typical in FD, but are also found in other subcellular organelles such as the ER, cell membrane, cytoplasm, and the nucleus [10,45].

Representative stainings of heart sections of each genotype are shown in Figure 5. As positive controls, we used respective kidney samples since the kidney has been reported to be the most affected organ in GLA^KO^ mice, and kidneys even of 10–14 weeks old mice have been reported to show detectable amounts of Gb3 depositions [27,46]. In line with this, only kidney tissue of GLA^KO^ mice revealed a clear typical staining of Gb3 positive granules located in the cytoplasm and nucleus as indicated by arrows, which was significantly stronger than in the kidney tissue of wild-type mice (Figure 5, upper row). The homogeneous signals in the cytoplasm in GLA^WT^ tissue are rather unspecific. In cardiac tissue of GLA^KO^ mice, Gb3 staining showed some granular signals in the cytoplasm (arrows) and very subtle signals in the nuclei. However, Gb3 staining of the cardiac tissue of GLA^WT^ and GLA^KO^ mice hardly revealed any difference between the genotypes. Thus, IHC staining in the cardiac tissue does not permit a definite and reliable detection of cardiac involvement in GLA^KO^ mice in contrast to CARS imaging (supported by computer-modulated analysis).

To assess Gb3 depositions in FD cardiac tissue and validate CARS microscopy results, we recorded the spectra of synthetic Gb3 and its main metabolite lyso-Gb3 (Figure 6). For both compounds, few distinct characteristic bands in the CH-stretching vibration region were determined. Precisely, it includes Raman bands of tissue components at ~2847 cm^−1^ corresponding to the CH_2_ symmetric stretching vibration of aliphatic lipid chains; the signals at ~2889 cm^−1^ corresponding to the CH_2_ asymmetric stretching vibration, and the signals at ~2940 cm^−1^ corresponding to the asymmetric CH_3_ stretching vibration of protein-associated compounds such as collagen, elastin and globular proteins [1,42]. A clear Raman shift in the CH vibrational region at 2847 cm^−1^ presented in the spectra of both substances refers to molecular vibrations of lipids. Additional measurements of both, Gb3 and lyso-Gb3, performed by Raman micro-spectroscopy, are available in the Appendix A. Both CARS and Raman reference spectra are in good agreement with the mean spectra of GLA^KO^ (see Figure 3 and Figure 4), where the lipid band is more dominant in GLA^KO^ than GLA^WT^ samples indicating a stronger lipid signal. Taken together, CARS microscopy has the potential for a reliable, fast, and robust differentiation between FD affected and non-affected heart samples.

## 3. Discussion

The study evaluated the potential of nonlinear spectroscopic imaging to identify cardiac involvement in FD and resulted in a tool for an automated prediction with a high mean sensitivity of FD in cardiac tissue samples. We used the α-Gal A knockout mice that display central characteristics of the FD phenotype, in particular a similar deposition pattern of Gb3 and its metabolite lyso-Gb3 as seen in a classical FD patient [20,22]. We implemented CARS microscopy to analyze the FD mouse model and proved that this method reliably predicts the genotype. For this, abundant multivariate data analysis was performed, including data clustering, biomarker analysis, and data classification. The reconstructed color maps of the heart sections based on the k-means clustering model demonstrated clear visual discrimination between the genotypes with a significant shift of the protein-lipid signal towards a lipid signal in GLA^KO^. Such a distinct differentiation between the genotypes was not seen using the classical IHC staining of Gb3. Also, the automated application of the PCA-LDA model achieved a clear separation (mean sensitivity of up to 96%) of mouse heart tissue of wild-type and FD mice. The study suggests CARS microscopy as a reliable method for estimating abnormal Gb3 accumulations in certain organs of FD even if they are not yet detectable by standard histological staining.

Several complex methods have been used so far in the diagnosis of cardiac involvement of FD: PET and cardiac MRI to detect myocardial fibrosis, echocardiography to monitor cardiac dysfunction, as well as metabolomics, proteomics, and electron microscopy to detect metabolic and structural changes [24,25,26,27]. But still, cardiac involvement of FD is underdiagnosed and presents the major cause of death in FD patients despite new and efficient therapeutic options. The herein described CARS microscopy is a novel, fast, and effective imaging tool that has the potential to support these well-established techniques. Mainly, due to its high sensitivity to the changes in lipid distributions, it allows a timely and reliable prediction of FD and has thus the potential to improve the patients’ prognosis. High spatial resolution, fast image acquisition, and the possibility to measure various types of sample (e.g., different tissue sections, in-vitro and ex-vivo cell samples or liquids) promotes CARS microscopy as a promising tool to support clinically well-established methods and enables new possibilities for clinical diagnosis, also concerning other diseases. Moreover, the sample, including its tissue structure, is maintained due to the stain-free technology making further sample investigation with other diagnostic modalities possible. In the clinic, it is of particular interest to be able to perform several analyses with cardiac biopsies since the extraction of cardiac biopsies is an invasive procedure, and the benefit of myocardial biopsy sampling needs to be weighed carefully against the risk. Moreover, our developed classification algorithms suggest that a single myocardial biopsy may be sufficient for diagnosing FD if compared to a respective database (Figure 4). Altogether, CARS microscopy is a method that can easily be combined with clinical standard tools (e.g., echocardiography, MRT, histology) and biochemical or biophysical methods (e.g., immune fluorescence, Raman, and IR spectroscopy, electron microscopy, etc.). The application of Raman micro-spectroscopy, a linear optical approach that we applied to isolated/synthesized Gb3 and lyso-Gb3 as shown in Appendix A, even allows distinguishing molecular information between these metabolites due to higher spectral resolution. However, the higher resolution of confocal Raman spectroscopy is associated with some disadvantages for the clinical practice such as slow measurement speed, which can take a few hours up to a day for comparable biopsy sections. Consecutively combined CARS and Raman spectroscopy would allow a fast tissue scan and genotype prediction by CARS and detailed biomarker information on the cellular level by Raman spectroscopy.

Further, the optical properties of the CARS microscope allow the combination of CARS with other image modalities like SHG or Stimulated Raman Scattering (SRS) microscopy for simultaneous pixel-wise measurements, which represents another advantage of CARS microscopy. Especially SRS delivers significant vibrational information with high sensitivity, which—in combination with CARS—could improve the performance of the machine learning algorithm making it more robust to outliers and improving the overall diagnostic procedure, not only for FD but also for other diagnostic applications. Nonlinear optical microscopy has already demonstrated its added value in the diagnosis of a brain tumor [47] and lung carcinoma [48], as well as the visualization of myocardial tissue remodeling [49]. Recently it was shown that the Raman-based characterization of tumor tissue provided an exact localization of cancer-infiltrated regions ex-vivo and in-vivo [50]. For tumor detection, even CARS-based endoscopy prototype systems are being developed [48,51], which may also be a new perspective for real-time spectroscopic imaging of vessels.

Clinically, an early diagnosis of FD with accumulations of Gb3 and its metabolites is of great importance to achieve the best therapeutic outcome. FD therapy is significantly less successful when cardiac remodeling is involved, or even progressive, irreversible organ damage is prevalent. Since small screening studies of patients with late-onset hypertrophic cardiomyopathy (=non-typical FD) have shown a severely delayed diagnosis of FD, the elaboration of novel methods for FD diagnosis (additional to standard methods such as the determination of enzyme activity), is of great relevance. Moreover, the detection and diagnosis of cardiac involvement in FD are of particular importance since cardiac involvement is asymptomatic for years and thus often diagnosed too late, causing the death of FD patients due to cardiac complications [12]. This is particularly true in women who, due to random X inactivation, often have normal enzyme activity and only selected organ involvement [10] and thus frequently display non-specific symptoms leading to a late if at all correct diagnosis [14]. On the other hand, even after the initial diagnosis, therapy decisions can be extremely troublesome in a considerable proportion of patients since disease courses can also be benign in certain variants and therapy costs are high [52]. Thus, in the effort to initiate FD-specific therapy at the earliest necessary time point while at the same time allowing to avoid unnecessary high therapy costs if the pathological substrate remains low, CARS microscopy might be particularly helpful not only in diagnosing but also in staging the disease and foreseeing clinical course. Thus, using the synergy of computer-aided bioanalytical techniques like multivariate data analysis and nonlinear optical imaging, we open up new prospects for applying biospectroscopy in the diagnosis of infiltrative diseases such as FD.

## 4. Materials and Methods

Fabry Disease Mouse Model. Two genotypes of male mice were investigated: GLA^WT^ mice with no α-Gal A gene deletion and GLA^KO^ mice that are hemizygous for the α-Gal A gene deletion [27]. FD is an inherited X-linked disease. Since hemizygous male mice display a typical human FD patient situation, male mice were used for the present study. Throughout the study, 20-week-old male mice in the C57BL/6J background were used and sacrificed by cervical dislocation for organ (heart and kidney) removal. Care of the animals was taken following the Committee on Animal Research of the regional government (Landesamt für Natur, Umwelt und Verbraucherschutz NRW, Recklinghausen, Germany), which reviewed and approved all experimental protocols (Az.84-02.04.2016.A404) according to the corresponding national legislation.

Adult Mouse Tissue Sections preparation for CARS Imaging. After organ removal, the fresh, untreated hearts were shock frozen in liquid nitrogen and stored at −80 °C. One day before tissue sectioning, hearts were placed in a −20 °C freezer to allow slow thawing. For tissue sectioning, the frozen organs were placed into the cryotome at −20 °C, and 6 μm thick tissue sections were prepared in the middle region of the left ventricle of the hearts and placed on microscope slides (Epredia™ SuperFrost Plus™ Adhesion slides 75 × 25 mm, Lauda Königshofen, Germany). All tissue sections were vacuum dried in a vacuum centrifuge for at least 30 min at room temperature (RT). Further, the samples were sealed with the 2-component silicon glue Panasil^®^ contact plus X-Light. The dried samples were stored at −80 °C until spectroscopic measurements were carried out on the CARS microscope. The measurements were performed within one and a half years, depending mainly on the age and the breeding schedule of the mice. The tissues of each litter were measured and analyzed in a blinded manner.

Immunohistochemical Staining of Heart and Kidney Tissue Sections. For localization of Gb3 deposits in adult mouse heart and kidney sections, immunohistochemical (IHC) staining was performed. Kidney tissue of GLA^KO^ and GLA^WT^ mice was used as a positive control since significant renal Gb3 deposition has been reported for GLA^KO^ in 20-week-old mice [21,27,46]. First, paraffin sections of 3–5 µm thickness were prepared. Before IHC staining, heat unmasking was performed in an EDTA buffer (pH 9.0) [52], followed by incubation with the primary monoclonal antibody anti-Gb3 from TCI-chemicals (A2506) at a dilution of 1/750 overnight at 4 °C. In the washing steps, the unbound antibody was washed off. A bridge antibody from Biomol (ASR2205) was incubated at a dilution of 1/1000 for 30 min at RT. This was followed by another washing step before the tissue slices were incubated with an anti-rabbit Polymer AP-conjugated from Enzo for 30 min at RT. Chromogen permanent red incubated for 2 × 10 min at RT completed the staining. Counterstaining was performed with hematoxylin [52,53].

Data Acquisition. CARS-SHG imaging was conducted with a Leica TCS SP8 CARS inverted microscope using Leica Application Suite X Software (Version 3.5.5.19976, Company: Leica Microsystems). The randomly selected regions of adult mouse heart sections were raster-scanned with a picosecond Nd:YVO4 OPO laser system (picoEmerald, APE, Berlin, Germany). The system is equipped with one baseline laser at a wavelength of 1064 nm resulting in a fixed and unmodulated Stokes beam and a tunable pump beam, where the laser could be tuned in the spectral range from 804.0 nm to 826.4 nm. The generated anti-Stokes scattering was detected in the forward direction (transmission). CARS-SHG hyper-spectral images were acquired via the above-mentioned automatic laser tuning with a step size of 0.7 nm generating 1024 × 1024 pixel size images at 33 equally distributed wavenumbers in the spectral range 2702.2 cm^−1^ to 3039.3 cm^−1^. A 40×/1.1 NA water immersion objective, 0.2 W laser intensity, and 10 Hz scan speed rate per wavenumber were utilized. In addition, the acquisition of a single spectral image took approximately 35 min. The data acquisition was performed at RT. The samples were taken from the −80 °C freezer and measured within 120–240 min. All samples were dried in a vacuum and warmed up to 28 °C inside the microscope chamber (temperature control at 28 °C) before starting the measurements. In addition, the heart tissue sections were imaged as tile scans at 816.5 nm, aiming to get a large overview of lipids distribution (characteristic CH_2_ stretching around 2850 cm^−1^) within the tissue. About four slides were prepared from each heart, and about 3–4 randomly selected areas from each slide were measured with the CARS microscope.

Reference Spectra Measurements of Gb3 and Lyso-Gb3. Standard biomarkers for FD (Gb3 and lyso-Gb3 in crystallized form) were measured on CARS and Raman microscopes [54]. CARS measurements were acquired similar to the above-described procedure. Gb3 and lyso-Gb3 were dried on the microscopic slide and placed into the CARS microscope warmed up to 28 °C. For the Raman measurements (see Appendix A), a WITec alpha300R Raman micro-spectroscope with laser radiation at 785 nm wavelength, a power of 200 mW (at the objective aperture), 30 s integration time, and 20 accumulations, and a 50×/0.75 NA dry objective were utilized. The WITec Software (Control FIVE 5.3.12.104) was used for the data acquisition. Data preprocessing and reconstruction were performed in R Studio (Version 3.4.2) and Python (Version 3.9.5) programming languages. The crystallized Gb3 (Ceramide trihexoside, Cat. No. 1067) and the lyso-Gb3 (lyso-Ceramide trihexoside, Cat. No. 1520) were ordered from the chemical manufacturer “Matreya LLC.”, State College, Pennsylvania, United States.

Multivariate Data Analysis and Machine Learning. The collected CARS hyperspectral images of adult mouse cardiac tissue sections were uploaded, preprocessed, and analyzed based on in-house written functions using the statistical programming language R (Version 3.4.2) and Matlab R2019a (Version 9.6.0.1072779). Each hyperspectral image in the acquired dataset is composed of 33 gray-scale images representing specific CARS wavenumbers, while the size of each gray-scale image is 1024 × 1024 pixels. For these hyperspectral datasets, the following described preprocessing sequence was performed along the image dimensions (xy) and the spectral dimension (Λ) of the hyperspectral data cube. First, the images were cleaned up from oversaturated pixels and down-sampled by a factor of two per axis/side. Along with the spectral dimension, baseline correction and vector normalization on the squared total intensity of all 33 wavenumbers were applied for each pixel spectrum. The preprocessed hyperspectral images obtained were utilized to explore spectral differences between knockout (GLA^KO^) and wild-type tissues (GLA^WT^). First, k-means clustering was performed to study the spectral similarity of the individual pixel spectra between both groups. From the obtained cluster maps, the relative amounts of pixels were calculated for each cluster, quantifying the overall spectral composition in each image.

After k-means clustering, the overall group separation was determined by machine learning via linear discriminant analysis (LDA) of the mean spectra of each hyperspectral image, which holds the information of all 512 × 512 pixels. The main principle of the LDA classification model is to find optimal linear combinations of variables that maximize the differences between our two genotypes but, on the other hand, minimize the differences within one group [7]. To facilitate the LDA classification process, the dimensionality of the mean spectra dataset was reduced via Principal Component Analysis (PCA) before the LDA. The resulting constructed model was then utilized to identify the GLA^KO^ tissues in the acquired datasets. In this context, similar PCA-LDA models were successfully applied for a supervised prediction of tumor tissue [55,56]. Here, the PCA-LDA models were conducted on the mean spectra of tissue sections without image scaling using an increasing number of principal components (PCs). The classification performance of this PCA-LDA model was checked using leave-one-mouse-out cross-validation (LOMO-CV) [44]. In this cross-validation strategy, the mean CARS spectra of one mouse image (one image of one tissue section of one mouse) are fixed as a test set, and the remaining mean spectra are utilized to build and train a PCA-LDA model. By repeating the same procedure using every mean spectrum of every image as a test set once, a generalization performance of the PCA-LDA model can be extracted. The classification model, in combination with the LOMO-CV, was applied to detect the reproducibility of the results and the significance between knockout and wild-type. Thus, one binary classification model for the FD mouse model was developed, which could classify GLA^WT^ against GLA^KO^. Besides, the minimal number of PCs to build the optimal LDA model was determined by testing several LDA models with different numbers of PCs (maximal numbers: 15) and comparing their mean sensitivities of the LOMO-CV, with the highest score being the optimal classification model.

## 5. Conclusions

In the present study, the application of CARS microscopy on cardiac tissue sections identified FD-affected biopsies with high mean sensitivity. This nonlinear spectroscopic approach allows (i) clear visual discrimination of affected and unaffected FD tissues; (ii) accurate identification of FD affected hearts based on machine-learning computer systems and comprehensive data analysis; (iii) the support of standard diagnostics since our study shows that the approach presented here allows early, reliable and efficient detection of FD-triggered alterations in affected organs. Ultimately, this method has the potential to help physicians initiate the needed therapy timely, to follow the patient’s response to the treatment, and to optimize cost-benefit considerations.

## Figures and Tables

**Figure 1 ijms-23-05345-f001:**
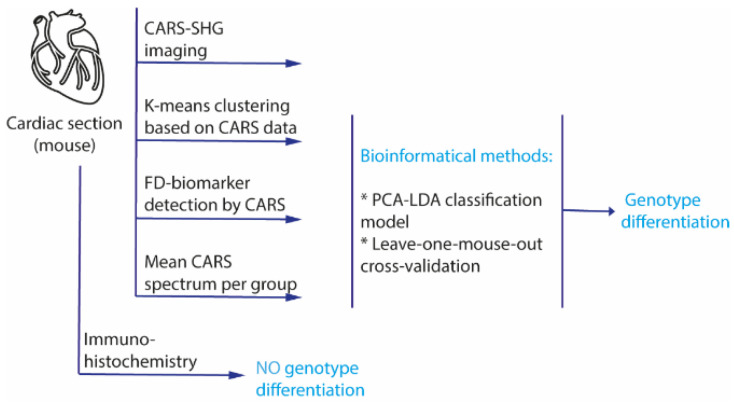
Schematic presentation of the workflow. Briefly, the unstained mouse heart sections were imaged using a CARS-SHG microscope. Multivariate data analysis was performed by data clustering, biomarker detection, and a classification approach based on the mean CARS spectra to bioanalytically differentiate between GLA^KO^ and GLA^WT^ mice. Additionally, classical immunohistochemical stainings (Gb3) of the cardiac sections of α-Gal A knockout (GLA^KO^) and wild-type (GLA^WT^) mice were performed.

**Figure 2 ijms-23-05345-f002:**
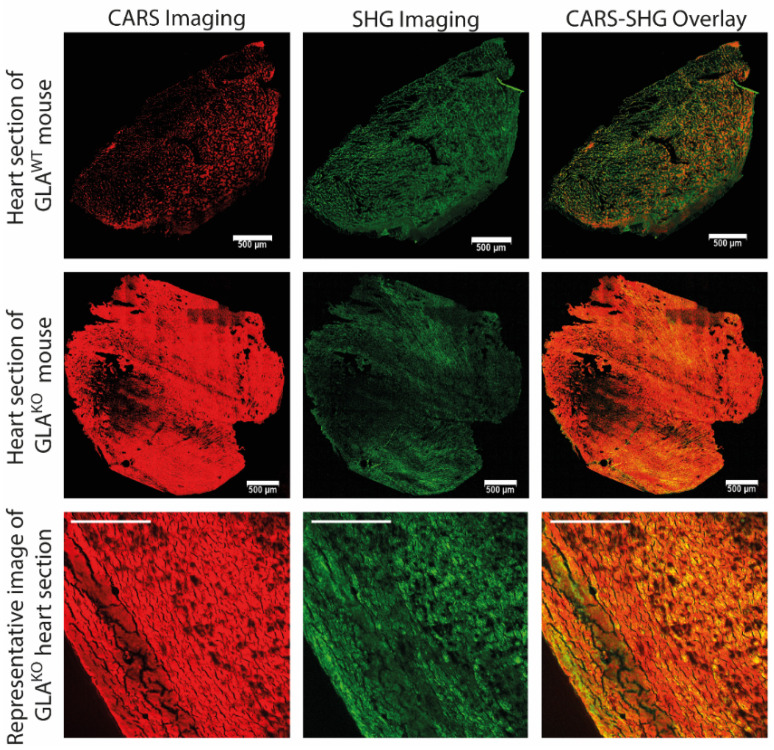
Nonlinear spectroscopy for the characterization of FD manifestation in heart sections of GLA^KO^ and GLA^WT^ mice, including coherent anti-Stokes Raman scattering (CARS) and second harmonic generation (SHG) imaging and their overlays. The CARS pump laser was tuned to 816.5 nm. CARS signal is depicted in red (left column), the SHG signal is depicted in green (middle column), and the overlay image represents the merged picture of the CARS and SHG signals (right column). The scale bars represent 500 µm for the overview images of the cardiac tissue section scans of both genotypes (upper and middle row) and 100 µm for the lower row that shows a representative magnification of a GLA^KO^ heart section. The overview pictures were stitched and merged automatically by the LasX software.

**Figure 3 ijms-23-05345-f003:**
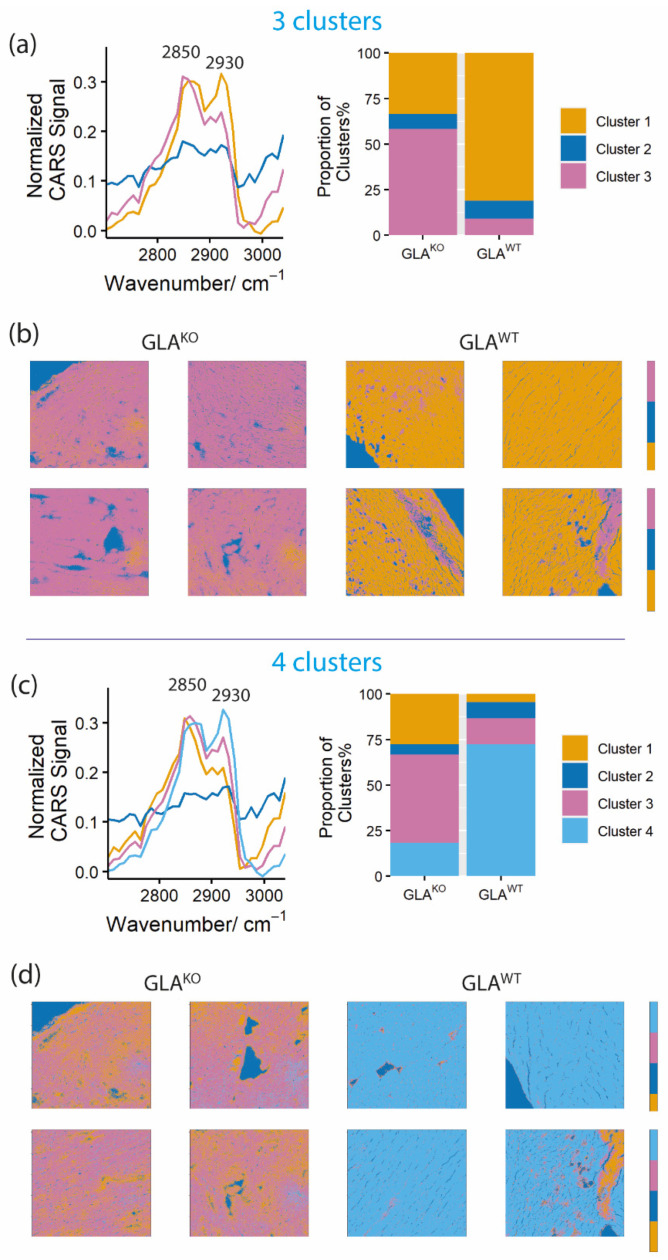
Clustering data analysis of CARS images of heart sections of the FD mouse model and wild-type mice allows the differentiation between knockout and wild-type mice (GLA^KO^, *n* = 35 vs. GLA^WT^, *n* = 20). K-means clustering is shown for three clusters (**a**,**b**) and four clusters (**c**,**d**). The clustering analysis was based on the ratio between lipids (CH_2_ stretch vibration at ~2850 cm^−1^) and proteins/lipids (CH_3_ stretch vibration at ~2940 cm^−^^1^); (**a**,**c**) show the mean spectra of three or four clusters, respectively, and the histograms, which display the distribution of the clusters in both genotypes in %; (**b**,**d**) show representative images of color maps for tissue sections of both genotypes, GLA^KO^ vs. GLA^WT^ (the image size is 500 µm × 500 µm).

**Figure 4 ijms-23-05345-f004:**
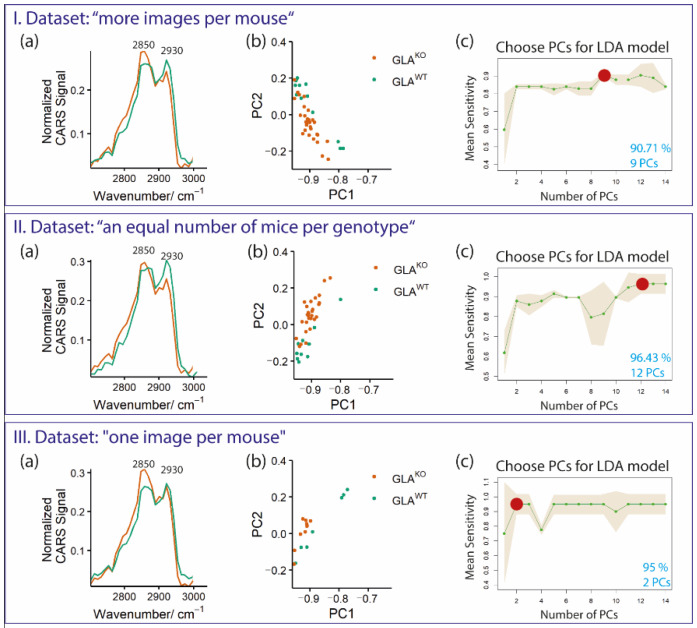
Statistical variations and data analysis of CARS images of heart tissue sections of a Fabry disease mouse model for three different datasets: (**I**). 20 GLA^WT^ vs. 35 GLA^KO^ (“more images per mouse”); (**II**). 10 GLA^WT^ vs. 28 GLA^KO^ (“an equal number of mice per genotype”); (**III**). 8 GLA^WT^ vs. 10 GLA^KO^ (“one image per mouse”). Shown is (**a**) a mean spectrum for each genotype; (**b**) PCA results (PC1 vs. PC2 for both genotypes); (**c**) PCA-LDA model based on leave-one-mouse-out cross-validation between both genotypes, demonstrating the mean sensitivity with standard deviation (depicted as a brown shadow; *y*-axis) for different numbers of PC included in the calculation (*x*-axis). The mean sensitivity was calculated as the average of all sensitivities for each genotype. The highest classification sensitivity in % and the number of PCs that are at least required to achieve this sensitivity are marked with a red dot, and the calculated value is indicated in blue.

**Figure 5 ijms-23-05345-f005:**
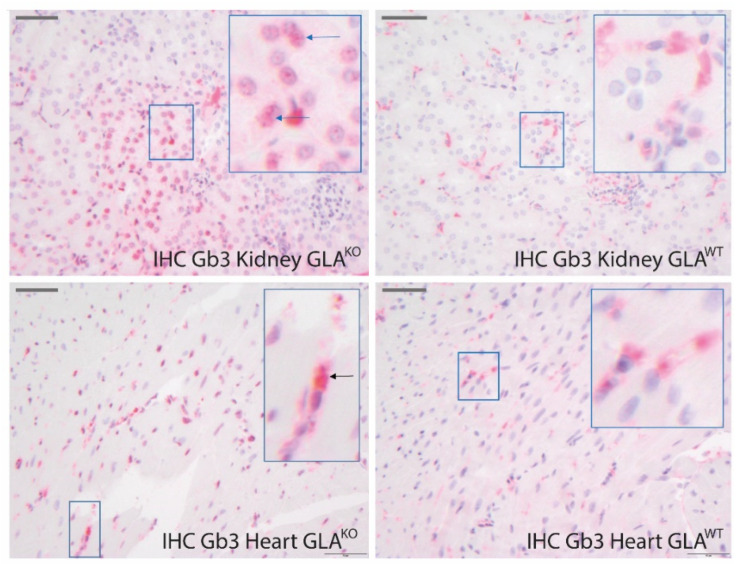
Immunohistochemical (IHC) staining of heart and kidney sections for Gb3 as a biomarker in FD. The organs were extracted from 20-week-old GLA^KO^ and healthy wild-type (GLA^WT^) mice. Scale bars represent 50 µm. Images were captured using Olympus microscope BX51 with 40× objective. Arrows show granular Gb3 positive stainings.

**Figure 6 ijms-23-05345-f006:**
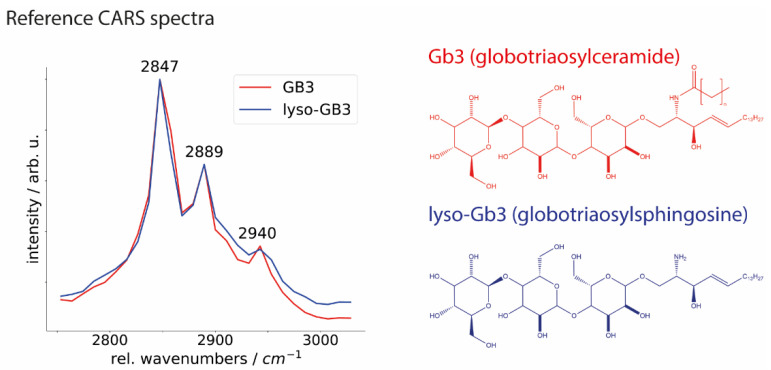
Reference spectra of globotriaosylceramide (Gb3, depicted in red) and its metabolite globotriaosylsphingosine (lyso-Gb3, depicted in blue) were recorded using a CARS microscope. The chemical structures of Gb3 and lyso-Gb3 are displayed in their respective colors. Relative (rel.) wavenumbers correspond to the measured compared to the laser baseline (785 nm) wavenumber.

## Data Availability

The datasets used and analyzed during the current study are available from the corresponding authors on reasonable request.

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
