# Peer review of "CARS Imaging Advances Early Diagnosis of Cardiac Manifestation of Fabry Disease"

_ijms, 2022, doi:10.3390/ijms23105345_

Round 1

Reviewer 1 Report

It is an surely interesting work. The results and sciences are well described. I basically recommend publication of this work in Int. J. Mol. Sci. I have some suggestions for revisions on general points. Please see below.

1) In some figures made through assembly of several panels, words in each panel are to small. Please increase font size and/or consider better arrangement of panels.

2) From visual items, this work looks like assemblies of spectra (and spectral methods) and bio samples that do not give impression of Molecular Science. Addition of chemical formulae (chemical structure scheme) may change this impression.

3) Although this is a nice research, descriptions in conclusive sections are too short and simple. more detailed descriptions on (i) comparisons with the other methods and (ii) future application perspective can be added.

Author Response

Dear Reviewer,

Thank you very much for the positive feedback and constructive comments. We included the suggested changes, discussion and figures. All changes are marked in grey within the manuscript. New version is attached.

In some figures made through assembly of several panels, words in each panel are too small. Please increase font size and/or consider better arrangement of panels.

We have increased the fond size of the figures and changed the general presentation of them.

From visual items, this work looks like assemblies of spectra (and spectral methods) and bio samples that do not give impression of Molecular Science. Addition of chemical formulae (chemical structure scheme) may change this impression.

Thank you for the important note! We included the chemical structures in Figure 3, which is now Figure 6. We improved and changed the figure legend and the manuscript text accordantly.

Although this is a nice research, descriptions in conclusive sections are too short and simple. More detailed descriptions on (i) comparisons with the other methods and (ii) future application perspective can be added.

Thank you for this helpful comment. We have included some more details, analyses and comparisons to other methods in the "Discussion" (lines 348-388).

With kind regards

Elen Tolstik and Kristina Lorenz

Reviewer 2 Report

In this manuscript, CARS images of the heart issues from mice have been analyzed via k-means clustering and PCA-LDA for diagnosis of Fabry disease (FD). Consequently, the authors have discriminated between not only knock-out and wild-typed mice, but also the genotypes spatially, as shown in Figs. 4 and 5. This method can be applied to the early diagnosis of FD. Thus, this is publishable in the journal. 

However, there are some comments as follows. 

Fig. 4: The similar spectrum in (a) and (c) should be shown by the similar color. The cluster 1 in Fig. 4(a) may correspond to the cluster 4 in FIg. 4(c). Moreover, the maps of the same tissues seem to be arranged at the different positions in (b) and (d). For example, the map in the upper right of (b) may correspond to that in the lower right of (d). 

Fig. 2: What is "rel."? Can be these omitted?

Line 109: reach -> rich ?

Lines 159, 161, 409: Λ -> λ

Line 289: Therefor -> Therefore

Author Response

Dear Reviewer,

Thank you for your positive feedback and the important comments. We have implemented your suggestions and have marked in grey the changes in the text.

However, there are some comments as follows. 

Fig. 4: The similar spectrum in (a) and (c) should be shown by the similar color. The cluster 1 in Fig. 4(a) may correspond to the cluster 4 in FIg. 4(c). Moreover, the maps of the same tissues seem to be arranged at the different positions in (b) and (d). For example, the map in the upper right of (b) may correspond to that in the lower right of (d). 

This is a very helpful suggestion: for a better visualization and interpretation of the color maps we arranged the images now as suggested in "similar" positions. We have changed Figure 4 (now Figure 3) accordingly. We also enlarged Figure 4 (now Figure 3) and added additional "3/4 clusters" to simplify the visualization for the reader.

Concerning the colors of the spectra: the presented results are from two different clusters, in our algorithm we had to avoid coloring the similar looking spectra in the same colors. Even if two spectra look contiguous, the differences in map distribution can be significant, because even one additional cluster can change the general sorting of data points by clusters and the interpretation of the results can be “visually” wrong. Therefore, we had to color the clusters based on random numbering by the algorithm.

Fig. 2: What is "rel."? Can be these omitted?

Here rel. (relative) corresponds to the measured wavenumbers compared to the wavenumber of the laser baseline (in this case 785 nm). For the better understanding we add now additional text (see lines 306-307).

Line 109: reach -> rich ?

Thank you for your comment. We have corrected the spelling.

Lines 159, 161, 409: Λ -> λ

Thank you for your note. Briefly to explain, with the capital lambda we refer to the wavenumber, whereas with the small lambda we refer to the wavelength (as provided by the Las X software of the CARS microscope). We have renamed it in the manuscript and in the Material and Methods part. We also introduced in the text “lambda” to make it easier for the reader (new line 169).

Line 289: Therefor -> Therefore

We have corrected the typo.

With kind regards

Elen Tolstik and Kristina Lorenz

Reviewer 3 Report

The manuscript should be of interest to the readers of the International Journal of Molecular Sciences. The level of English is sufficient; however, some sentences are not easy to understand and the manuscript is peppered with grammar mistakes and typos. The manuscript is not always clear and at times it can be confusing. The following points should be taken into consideration.

Abstract

Line 35: What does "the chosen time point" mean? What time is that?

Line 36: Why is kidney tissue of interest when the focus is on cardiac tissue?

Introduction

As a general comment, this section is a bit incoherent. There is a lot of information but it is disorganised and does not follow a logic structure.

Line 46: Sentence "hence, new alternatives ..." appears to be incomplete; the verb is missing

Line 53: What does "other cellular compartments" refer to? Which compartments are these?

Line 54: What does "extracellularly" refer to?

Line 54: Sentence "FD is considered ..." seems to be a repetition of what already said in line 49

Line 56: Sentence "cardiac involvement ..." seems to be a repetition of what already said in line 49

Line 75: Saying "if FD is diagnosed" sounds obvious. Further clarification is needed

Line 88: What does "chemical particularity" mean?

Line 96: What is a "reasonable speed"? Reasonable compared to what? Further clarification is needed

Results

Figure 1: It is a confusing figure because it is not entirely clear what it tries to show (there is not much explanation in the caption or in the text). Since its content is confusing, it is unclear whether it should be moved to the section Materials and Methods. It would be better to divide the figure into multiple separate images. Furthermore, the images are too small and it is not always clear what the reader is looking at

Lines 123 - 133: This paragraph should be moved to the section Materials and Methods because it describes how the CARS signals were acquired

Line 136: It is unclear why the CARS signals had to be validated by standard Raman spectroscopy. After all, CARS is used instead of Raman so why it is necessary to collect Raman spectra?

Figure 2: It would be better to use different types of curves (e.g. dashed curves and dotted curves) instead of curves of different colours

Line 154: Sentence "As mentioned before ..." is redundant and should be removed

Lines 155 - 163: These sentences should be moved to the section Materials and Methods because they do not describe results but only how CARS measurements were carried out

Figure 3: It is unclear what these images represent and there is not much explanation in the caption or the text. Do they represent the same area? How can they be compared? The scale bars are not easily spotted

Lines 170 - 183: These sentence should be moved after Figure 4

Line 172: What "cluster" is that?

Line 182: To which figure sentence "of interest, CARS images ..." refer to?

Figure 4: Unclear collage of images. Images are too small and the labels in the images in (b) and (d) are not readable because their size is too small. What are the CARS peaks normalised to? The CARS peaks should be labelled. What do "cluster 1", "cluster 2", etc. refer to? How were the histograms in (a) and (c) obtained?

Lines 191 - 203: These sentences should be moved before Figure 4

Lines 199 - 205: These sentences should be moved to section Discussion because they interpret the results obtained

Lines 206 -  215: These sentences should be moved to section Materials and Methods because they do not describe results but only how PCA-LDA analysis was carried out

Line 216: What does "various selected data sets" mean? How were they selected?

Lines 230 - 233: These sentences are unclear and confusing. They are written in a sort of caption style rather than as proper text

Figure 5: CARS peaks should be added to the spectra. The separation between the two clusters in PC1-PC2 plots is not clear as the clusters overlap and are not clearly separated. How was the "mean sensitivity" calculated? What does "choose PCs for LDA model" mean? The legend in (c) is too small. What does the gray area in (c) refer to? How was it obtained?

Line 233: The sentence "were different from GLAWT ..." does not describe sufficienty well what Fig 5c shows because the two clusters are not well separated and overlap. In addition, there are outliers that are unaccounted for (i.e. it is not clearly explained what they are and mean)

Line 235: The sentence "corresponding signals in Figure 4" is unclear because it is not clear which peaks are considered here

Line 238: What does "slight separation" refer to, since the clusters in Fig. 5b are not separated? Clarify

Line 246: How was the mean sensitivity calculated?

Line 249: "Marked as red dots"; these red dots are very difficult to spot in Fig 5(c)

Lines 255 -  258: These sentences should be moved to the section Discussion because they do not describe any results but they interpret them

Figure 6: What is the size of the images (including the insets)? Why are kidney tissues studied when the focus is on cardiac tissues?

Discussion

As a general comment, this section is not a Discussion because there is no interpretation as such of the results presented in section Results. As it stands the current section Discussion is another Introduction. The results have not been properly discussed in this section; there are only some very brief discussions of the results in the section Results. It would be worth considering to combine the two sections and have a section "Results and Discussion"

Line 287: What "high accuracy" refer to? Sensitivity was calculated but not accuracy

Line 289: Change "therefor" into "therefore"

Line 298: What does "murine heart tissue of WT" refer to?

Materials and Methods

As a general comment, this section should be moved before the section Results. Information on how many samples were used should be added. It should also be clearly stated whether the experiments were done over multiple days. The environmental conditions in which experiments were conducted should also be added. Was an environmental chamber used when performing CARS measurements? How long was the acquisition of each of the CARS images/spectra?

Line 352: Is the "organ" the heart? Clarify

Line 357: Is "organ" refer to the heart?

Line 358: Which "organs"? The heart? What other organs?

Line 362: Why were the kidneys considered when the focus of the study was on heart tissue?

Line 380: What does "applying" mean? Does it mean thar the CARS instrument was motorised and controlled via a software? Clarify

Line 381: What does "selected regions" refer to? Which regions are these? How were they selected?

Line 382: What type of laser was used? What was its power?

Line 383: What type of laser was used? What was its power? What was the function of each of the two lasers? Clarify

Line 385: How many "hyper-spectral" images were acquired?

Line 386: What does "33 wavenumbers" mean? Does it mean that 33 different wavenumbers were chosen? Clarify

Line 387: Was "0.2W" the laser power at the sample? Which laser?

Line 387: Was "10Hz" the speed of the scan? Clarify

Line 391: What is a "microscopic box"? Is this an environmental chamber?

Line 393: What type of laser (wavelenght and power) was used for Raman experiments?

Line 396: Is "200mW" the laser power at the sample?

Line 397: How long was the acquisition time of the Raman spectra?

Line 398: What version of R studio and Python were used?

Line 402: It is unclear what sort of machine learning was carried out

Line 405: Why was Matlab used when earlier it was said that Python was used? Clarify

Line 406: What is the range of the "specific CARS wavenumbers"?

Line 408: What does "pipeline" mean in this context?

Line 411: Why were the images "down-sampled by a factor of two"?

Line 411: Explain "some samples". Which samples? Why not all of the samples?

Line 428: Explain "mean spectra". Mean over how many spectra? Were the spectra similar to each other to allow for their mean to be calculated?

Line 429: How many PCs were taken into account? Why?

Author Response

Dear Reviewer,

Thank you very much for your extremely helpful comments. We have reorganized several parts of the manuscript and included several more detailed descriptions and thus hope that our data are now more clearly presented. All changes are marked in gray within the manuscript. Our answers are in a separate file to make them easier for you to understand.

With kind regards

Elen Tolstik and Kristina Lorenz

Round 2

Reviewer 3 Report

Thank you to the authors for taking on board the comments and acting on them.